# SELFIES-TED : A Robust Transformer Model for Molecular Representation using SELFIES

## Abstract

Large-scale molecular representation methods have revolutionized applications in material science, such as drug discovery, chemical modeling, and material design. With the rise of transformers, models now learn representations directly from molecular structures. In this paper, we introduce SELFIES-TED, a transformer-based model designed for molecular representation using SELFIES, a more robust, unambiguous method for encoding molecules compared to traditional SMILES strings. By leveraging the robustness of SELFIES and the power of the transformer encoder-decoder architecture, SELFIES-TED effectively captures the intricate relationships between molecular structures and their properties. Having pretrained with 1 billion molecule samples, our model demonstrates improved performance on molecular property prediction tasks across various benchmarks, showcasing its generalizability and robustness. Additionally, we explore the latent space of SELFIES-TED, revealing valuable insights that enhance its capabilities in both molecule property prediction and molecule generation tasks, opening new avenues for innovation in molecular design.

## 1 Introduction

Large-scale molecular representation methods are shown to be useful in various material science applications, such as virtual screening, drug discovery, chemical modeling, material design, and molecular dynamics simulations. With the progress in deep learning, numerous models have been developed to derive representations directly from molecular structures. Recently, transformer-based molecular representations have gained prominence in material informatics, offering significant potential for advancements in drug discovery, materials science, and related fields. Recent works Chithrananda et al. (2020); Bagal et al. (2021); Ross et al. (2022); Chilingaryan et al. (2022); Yüksel et al. (2023) have demonstrated the capability of transformer models in capturing complex relationships and patterns within molecular data with the help of attention mechanisms. Most of these works are based on SMILES (Simplified Molecular Input Line Entry System) (Weininger (1988)). However, one of the drawbacks of SMILES is that it does not guarantee syntactic and semantic validity of the molecule (Krenn et al. (2020)), thus leading to a possibility of learning invalid representations. SELFIES (SELF-referencing Embedded Strings) is another molecular string representation that was introduced by Krenn et al. (2020) to overcome the drawbacks of SMILES. Yüksel et al. (2023) has demonstrated the effectiveness of a transformer encoder model trained with SELFIES. However, in addition to achieving high accuracy predictions of molecular properties, a key objective within computational material informatics is to devise novel and functional molecules. But most existing transformer models for material informatics are encoder-only models, which are not capable of generating new molecules.

In this paper, we introduce SELFIES-TED, a transformer-based model capable of capturing intricate molecular relationships and interactions. Unlike most existing works that utilize encoder-only models, we propose an encoder-decoder model based on BART (Bidirectional and Auto-Regressive Transformers) (Lewis et al. (2019)). This model not only efficiently learns molecular representations but is also capable of auto-regressively generating new molecules from these representations. This capability is particularly impactful for novel molecule design and generation, facilitating efficient and effective analysis and manipulation of molecular data. The main contributions of this paper are:

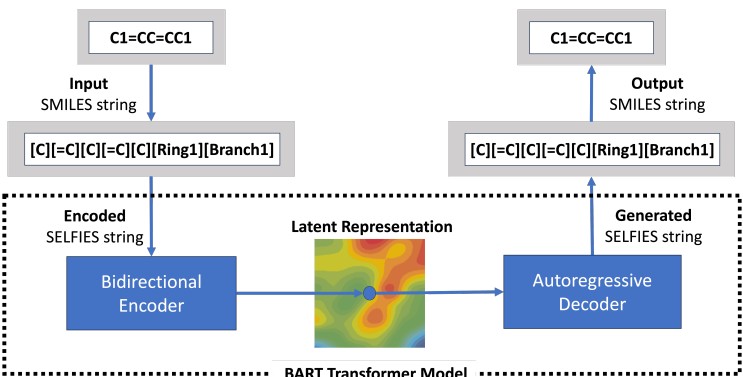

Figure 1: Model architecture

- We present a robust transformer-based model for molecular representation pretrained using 1B molecules represented by SELFIES strings, and demonstrate its effectiveness through evaluations on standard benchmarks.

- We perform an in-depth analysis of the model's latent space, providing insights into the representation of molecular features and introduce a multi-view representation approach to enhance the quality and diversity of these representations.

- We demonstrate how the learned representations can be applied to molecule generation tasks, showing that our model is effective at generating novel molecules and improving upon existing ones when conditioned upon desired properties.

## 2 PROPOSED SELFIES-TED MODEL

The proposed SELFIES-TED model is an encoder-decoder architecture derived from the BART (Bidirectional Auto-Regressive Transformer) model Lewis et al. (2019). The encoder processes the input token sequence bidirectionally, while the decoder generates the sequence autoregressively. The SELFIES-TED model is trained using SELFIES as it provides a more concise and interpretable representation, making it suitable for machine learning applications where compactness and generalization are important (Krenn et al. (2020)). We present two variants of the pre-trained models, namely SELFIES-TED$_{small}$ and SELFIES-TED$_{large}$. The SELFIES-TED$_{small}$ model is a 2.2M parameter model with 2 encoder-decoder layers and 4 attention heads, pretrained with 8B samples from ZINC-22 dataset (Tingle et al. (2023)). The SELFIES-TED$_{large}$ model is a 358M parameter model with 12 encoder-decoder layers and 16 attention heads, pretrained with 1B samples curated from a mixture of ZINC-22 (Tingle et al. (2023)) and PubChem (Kim et al. (2016)) datasets. The models were trained on NVIDIA V100 16GB GPUs.

**Tokenization** : The SELFIES-TED model employs a word-level tokenization scheme tailored to the structure of SELFIES. In this scheme, each symbol within a SELFIES string, enclosed in square brackets (e.g., [C], [=O], [Branch1]), is treated as an individual token. These tokens encapsulate fundamental molecular features such as atoms, bonds or branching points, providing a structured and interpretable representation of molecular data. The SELFIES-TED models are trained using the ZINC-22 and PubChem datasets, which primarily represent molecules in SMILES notation. We convert these SMILES strings to SELFIES using the SELFIES API (Krenn et al. (2020)). The SELF-IES API encodes the SMILES string into a SELFIES string where each atom or bond is represented by symbols enclosed in [ ], which are then tokenized using the word level tokenization scheme where each symbol or bond in [ ] is treated as a word. The SELFIES-TED$_{small}$ model, trained exclusively on the ZINC-22 dataset, has a vocabulary size of 173. This is because ZINC-22 primarily contains small molecules with limited token diversity. In contrast, the SELFIES-TED$_{large}$ model trained on both ZINC-22 and PubChem encounters a significantly broader range of molecular structures, resulting in a much larger vocabulary size of 3160.

**Model Pre-training** : During pre-training, the model is trained with a denoising objective, where 15% of the tokens in the input sequence are randomly masked. The encoder processes this corrupted

input sequence ($X_{\text{corrupt}}$), and the decoder is trained to autoregressively predict the next token in the original sequence ($Y$), conditioned on the corrupted sequence and the previously decoded tokens ($Y_{<t}$). This ensures that the model attempts to learn the semantic structure of the sequence as the decoder learns to recover the original sequence by predicting each token in the correct order. The objective function for training is defined as:

$$\mathcal{L}_{\text{denoise}} = -\sum_{t=1}^{T} \log P(Y_t | Y_{<t}, X_{\text{corrupt}}; \theta)$$

where, $Y_t$ is the $t$-th token in the original sequence ($Y$), $Y_{<t}$ is the sequence of tokens preceding $t$ in the target sequence, $X_{\text{corrupt}}$ is the corrupted input sequence with random masking, $\theta$ is the model parameters, and $P(Y_t | Y_{<t}, X_{\text{corrupt}}; \theta)$ is the model's predicted probability of token $Y_t$ given the context.

**Downstream Task Fine-tuning** : For downstream tasks such as property prediction, we use the encoder output a.k.a. latent representation, averaged over the sequence length as input feature to train a simple downstream model to predict properties. The latent representation vector is of dimensions 256 and 1024 for the SELFIES-TED$_{\text{small}}$ and SELFIES-TED$_{\text{large}}$ models, respectively.

Figure 1 illustrates the pre-training model architecture. We hypothesize that the encoder-decoder structure of the SELFIES-TED model, combined with the denoising objective, provides better molecular representations. Moreover, training on SELFIES instead of SMILES ensures that the encoder output represents only valid molecules, enhancing the robustness of the molecular representations which are used for downstream tasks such as property prediction.

## 3 PROPOSED MULTI-VIEW REPRESENTATION

In this section we further extend our study on the latent representation and propose a Multi-View Representation (MVR) approach to enhance the quality of the learned representations.

While foundation models have shown great promise in materials science and chemistry, they face a significant challenge: the limited availability of large, diverse datasets, especially in the downstream tasks such as property prediction. In contrast to the vast text corpora used to train large language models (LLMs), datasets in materials science and chemistry often contain only a few hundred samples. This data scarcity hampers the ability to train models that generalize well on unseen molecular structures, especially for tasks requiring high-quality latent representations.

One common approach to addressing this issue is SMILES enumeration (Bjerrum (2017)), a data augmentation technique that generates multiple valid SMILES representations for the same molecule by varying the starting atom and traversal order within the molecular graph. The same enumeration can be extended to SELFIES strings too. Figure 2 illustrates an example of a molecule represented by several different SMILES/SELFIES strings. While this method increases the dataset sample size, it does not necessarily enhance the quality or expressiveness of the latent space learned by the model. Simply adding more samples might improve training performance, but it does not guarantee that the learned representations effectively captures the molecular properties.

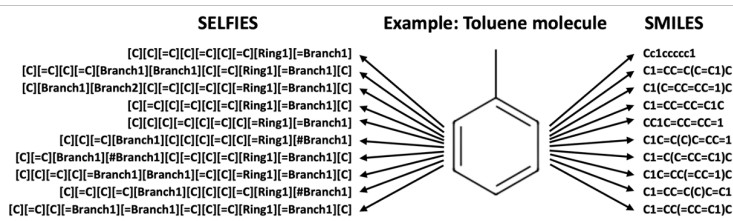

Figure 2: Example of SMILES/SELFIES enumeration where a single molecule can be represented in multiple forms

Driven by the need to understand how the latent representations of enumerated strings relate to each other, given that they represent the same underlying molecule, we further extend our study

on representations of the enumerated strings. We conducted a preliminary analysis by selecting 10 random molecules from the MoleculeNet (Wu et al. (2018)) BACE dataset and generated 100 alternative SMILES strings for each. We then extracted the latent representations using our proposed SELFIES-TED model that was pretrained on 1 billion samples, and visualized the representation using t-SNE visualization (Van der Maaten & Hinton (2008)). As shown in Figure 3, clear clusters emerged, where each cluster corresponds to a molecule and its alternative representations. The latent representations of the enumerated SMILES/SELFIES form a cloud, indicating that these alternate forms cluster together and can be treated as different views of the same molecule, each conveying a different aspect of the same underlying molecule. Notably, some clusters, such as those for molecule pairs $\{6, 8\}$ and $\{3, 5\}$, exhibit overlaps. Upon examining their molecular structures, we can observe that these molecules share significant structural similarities, likely causing their representations to align closely in the latent space.

Building on these observations, we propose a novel framework called Multi-View Representation (MVR) to enhance the expressiveness of molecular latent representations for molecular modeling.

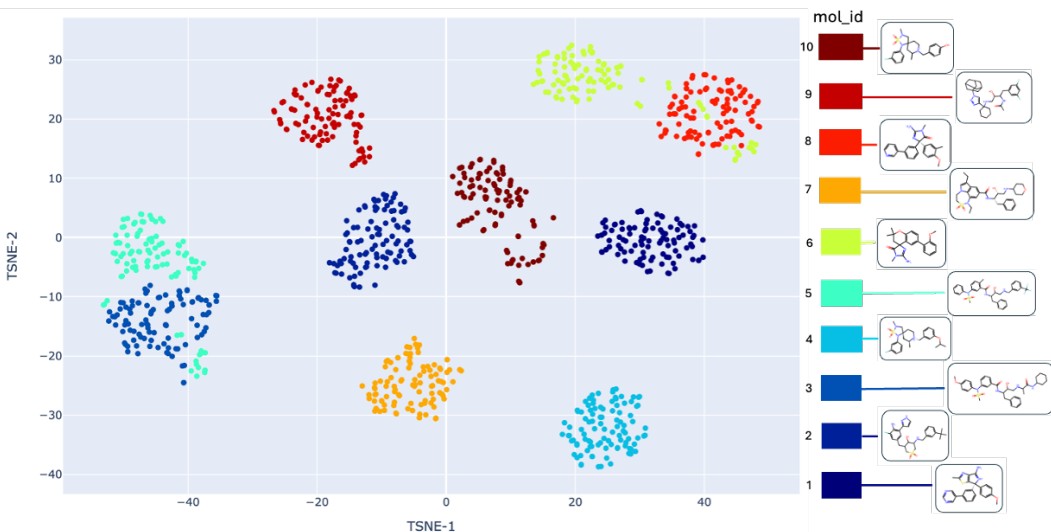

Figure 3: T-SNE plot of the latent representation 10 different molecules and their enumerated forms

The core idea is to generate multiple latent representations for the same molecule, each capturing distinct features or "views" of the molecule. By systematically selecting and combining these features, we create a more comprehensive and enriched latent vector representation. This approach is expected to improve the quality of the latent space representation, consequently improving the performance in downstream tasks such as molecular property prediction. The schematic of the proposed MVR framework is illustrated in Figure 4. The proposed framework operates through three main steps:

- Generating multiple string representations: Obtain $k$ different SMILES or SELFIES strings for the same molecule, including canonical and non-canonical variants. These alternate string representations provide different "views" of the molecule's structure.

- Extracting Latent Representations: For each generated string, we use a pretrained model (Eg. SELFIES-TED model) to obtain its latent representation. Each latent representation is hypothesized to capture different aspects or "views" of the molecule's structure and properties.

- Selecting and Combining Latent Representations: To create an enriched representation, a greedy selection process is used to identify the most informative latent vectors. These selected vectors are concatenated to form a unified, comprehensive latent representation that leverages the diversity of the alternate views.

The final enriched feature vector is fed into a downstream model to make predictions. By leveraging multiple views of the molecule, this approach is expected to enhance molecular modeling by capturing a broader spectrum of molecular features from different latent views, ultimately improving performance in various cheminformatics tasks.

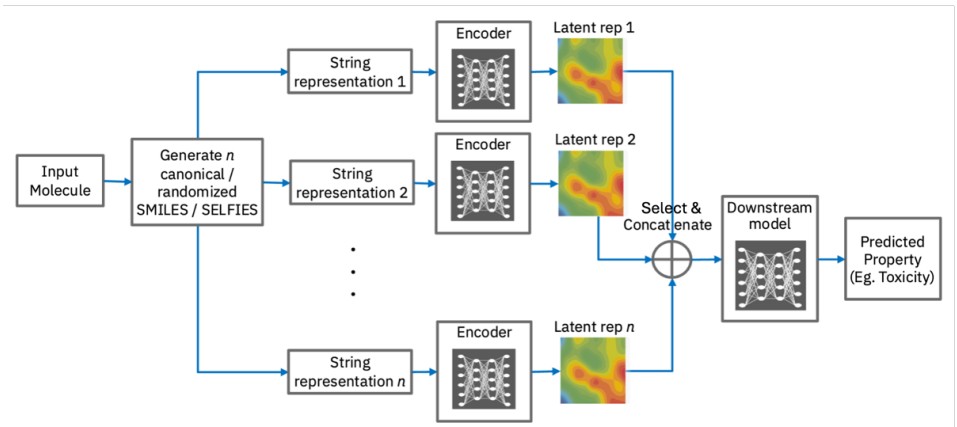

Figure 4: Proposed Multi-View Representation

## 4 RESULTS AND DISCUSSIONS

We evaluate the effectiveness of our proposed SELFIES-TED model on both molecular property prediction tasks and molecule generation tasks. For the molecular property prediction tasks, we conducted evaluations using a comprehensive set of 10 distinct benchmark datasets sourced from MoleculeNet (Wu et al. (2018)). The details of the benchmarks used are illustrated in Table 1. We evaluate 6 datasets for the classification task and 4 datasets for regression tasks. To ensure a robust and unbiased assessment, we maintained consistency with the MoleculeNet benchmark by adopting identical train/validation/test splits for all tasks (Wu et al. (2018)). We compare the performance of the proposed SELFIES-TED model with various graph-based and text-based models. We also conducted a brief evaluation of the proposed multi-view representation approach to show that the additional information provided by alternate representations of a molecule can improve the prediction results. In addition we also evaluate the capability of the proposed SELFIES-TED model in molecule generation tasks and compare its results with existing molecular generative models.

| Dataset | Description | Task | #Samples | Metric |
|---------|-------------|------|----------|--------|
| BACE | Binary labels on $\beta$-secretase 1 (BACE1) binding properties | 1 | 1,513 | ROC-AUC |
| ClinTox | Binary labels on clinical toxicity data on FDA-approved drugs | 1 | 1,478 | ROC-AUC |
| BBBP | Binary labels on blood–brain barrier permeability | 1 | 2,039 | ROC-AUC |
| HIV | Binary labels on the ability to inhibit HIV replication | 1 | 41,127 | ROC-AUC |
| SIDER | Drug side effect classification for 27 types of adverse effects | 27 | 1,427 | ROC-AUC |
| Tox21 | Qualitative toxicity measurements on 12 targets | 12 | 7,831 | ROC-AUC |
| Esol | Water solubility prediction of small molecules | 1 | 1,128 | RMSE |
| Lipophilicity | Prediction of octanol-water partition coefficient (logD) | 1 | 4,200 | RMSE |
| Freesolv | Hydration free energy of small molecules in water | 1 | 642 | RMSE |
| QM9 | Quantum mechanics properties of DFT-modelled small molecules | 12 | 133,885 | MAE |

Table 1: Description of the benchmark datasets used in the evaluation of the proposed model.

### 4.1 MOLECULAR PROPERTY PREDICTION TASKS

We evaluated the SELFIES-TED models on 10 benchmarks from MoleculeNet (Wu et al. (2018)). These tasks include four binary classification tasks using BACE, ClinTox, BBBP and HIV datasets, two multi-label classification task using SIDER and Tox21 datasets, and three single-output regression tasks using the esol, freesolv and lipophilicity and a multi-output regression task using the QM9 dataset. For the evaluations, the model weights are frozen and not finetuned in the downstream tasks and the molecular embeddings generated by the SELFIES-TED models are used as input features to the downstream model. First we evaluate the model performance on classification tasks. We use XGBoost (Chen & Guestrin (2016)) as the downstream task model and Optuna (Akiba et al. (2019)) for hyperparameter tuning. The following key hyperparameters were considered during the tuning process: the number of boosting rounds ranging from 1000 to 10000 and the booster type , chosen from gbtree, gblinear, and dart. Regularization parameters such as $\lambda$ and $\alpha$ were tuned within a logarithmic range of $10^{-8}$ to 1.0. The maximum tree depth was optimized between 1 and 12, while the learning rate ($\eta$) and gamma ($\gamma$) were both explored in the range of $10^{-8}$ to 1.0. Additional hyperparameters included the growth policy, selected from depthwise and lossguide, and subsample ratio

| Model | BBBP ↑ | ClinTox ↑ | HIV ↑ | BACE ↑ | SIDER ↑ | Tox21 ↑ |
|---|---|---|---|---|---|---|
| Morgan Fingerprint | 93.0 | 82.8 | 80.0 | 88.5 | 68.2 | 66.8 |
| RF (Ross et al. (2022)) | 71.4 | 71.3 | 78.1 | 86.7 | 68.4 | 76.9 |
| SVM (Ross et al. (2022)) | 72.9 | 66.9 | 79.2 | 86.2 | 68.2 | 81.8 |
| MGCN (Lu et al. (2019)) | 85.0 | 63.4 | 73.8 | 73.4 | 55.2 | 70.7 |
| D-MPNN (Yang et al. (2019)) | 71.2 | 90.5 | 75.0 | 85.3 | 63.2 | 68.9 |
| DimeNet (Gasteiger et al. (2020)) | - | 76.0 | - | - | 61.5 | 78.0 |
| Hu, et al. (Hu et al. (2019)) | 70.8 | 78.9 | 80.2 | 85.9 | 65.2 | 78.7 |
| N-Gram (Liu et al. (2019)) | 91.2 | 85.5 | 83.0 | 87.6 | 63.2 | 76.9 |
| MolCLR (Wang et al. (2022)) | 73.6 | 93.2 | 80.6 | 89.0 | 68.0 | 79.8 |
| GraphMVP (Liu et al. (2021)) | 72.4 | 77.5 | 77.0 | 81.2 | 63.9 | 74.4 |
| GeomGCL (Liu et al. (2021)) | - | 91.9 | - | - | 64.8 | 85.0 |
| GEM (Fang et al. (2022)) | 72.4 | 90.1 | 80.6 | 85.6 | 67.2 | 78.1 |
| ChemBerta (Chithrananda et al. (2020)) | 64.3 | 73.3 | 62.2 | 79.9 | - | - |
| ChemBerta2 (Ahmad et al. (2022)) | 71.94 | 90.7 | - | 85.1 | - | - |
| Galatica 30B (Taylor et al. (2022)) | 59.6 | 82.2 | 75.9 | 72.7 | 61.3 | 68.5 |
| Galatica 120B (Taylor et al. (2022)) | 66.1 | 82.6 | 74.5 | 61.7 | 63.2 | 68.9 |
| Uni-Mol (Zhou et al. (2023)) | 72.9 | 91.9 | 80.8 | 85.7 | 65.9 | 79.6 |
| MoLFormer-XL (Ross et al. (2022)) | 93.7 | 94.8 | 82.2 | 88.2 | 69.0 | **84.7** |
| SELFormer (Yüksel et al. (2023)) | 90.2 | - | 68.1 | 83.2 | **74.5** | 65.3 |
| MolGen-large (Fang et al. (2024)) | 92.5 | 74.4 | 75.6 | 85.9 | 62.2 | 73.8 |
| **SELFIES-TED**$_{small}$ | 92.6 | 88.3 | 74.2 | 87.0 | 62.4 | 75.1 |
| **SELFIES-TED**$_{large}$ | **95.2** | **96.9** | **83.0** | **89.3** | 65.0 | 76.5 |

Table 2: Results of the evaluation on classification tasks of MoleculeNet benchmark datasets

and column sampling ratio, both varied between 0.05 and 1.0. Each classification task underwent independent hyperparameter optimization to ensure the downstream model was best tailored to the embeddings generated by the SELFIES-TED models. Performance was evaluated using the ROC-AUC metric, with results reported for the optimal hyperparameter configurations. Table 2 presents the performance of the SELFIES-TED models compared to other molecular graph-based, geometry-based models and molecular string-based models. ChemBERTa, Galatica, Uni-Mol and MolFormer are trained on SMILES representations, while SELFormer, MolGen and the proposed SELFIES-TED models are trained on SELFIES representations. As shown in Table 2, the SELFIES-TED$_{large}$ model outperforms the other models in four out of six tasks. Meanwhile, SELFIES-TED$_{small}$ model shows competitive performance compared to the other graph and text based models.

| Model | ESOL ↓ | FreeSolv ↓ | Lipophilicity ↓ |
|---|---|---|---|
| Morgan Fingerprint | 0.769 | 1.756 | 0.691 |
| D-MPNN (Yang et al. (2019)) | 1.050 | 2.082 | 0.683 |
| Hu et al. (Hu et al. (2019)) | 1.220 | 2.830 | 0.740 |
| MGCN (Lu et al. (2019)) | 1.270 | 3.350 | 1.110 |
| GEM (Fang et al. (2022)) | 0.798 | 1.877 | 0.660 |
| SchNet (Schütt et al. (2017)) | 1.050 | 3.220 | 0.910 |
| KPGT (Li et al. (2022)) | 0.803 | 2.121 | **0.600** |
| GraphMVP-C (Liu et al. (2021)) | 1.029 | - | 0.681 |
| GCN (Kipf & Welling (2016)) | 1.430 | 2.870 | 0.850 |
| GIN (Xu et al. (2018)) | 1.450 | 2.760 | 0.850 |
| MolCLR (Wang et al. (2022)) | 1.110 | 2.200 | 0.650 |
| ChemBERTa-2 (Ahmad et al. (2022)) | - | - | 0.986 |
| MolFormer (Ross et al. (2022)) | 0.755 | 2.022 | 0.840 |
| SELFformer (Yüksel et al. (2023)) | 0.682 | 2.797 | 0.735 |
| MolGen-large (Fang et al. (2024)) | 0.499 | 1.514 | 0.704 |
| **SELFIES-TED**$_{small}$ | 0.506 | 1.779 | 0.822 |
| **SELFIES-TED**$_{large}$ | **0.454** | **1.147** | 0.672 |

Table 3: Results of the evaluation on regression tasks of MoleculeNet benchmark datasets

We also evaluate the performance of the models on 3 regression tasks, the results of which are presented in Table 3. The SELFIES-TED$_{large}$ model outperforms the other models in two out of three tasks. SELFIES-TED$_{small}$ still shows better performance compared to the other text-based models. We further analyze the performance of our best performing SELFIES-TED$_{large}$ model on the QM9 dataset, comparing its performance with Molformer-XL and ChemBERTa, both text-based models trained on SMILES. Note that the weights of all the models are frozen and not fine-tuned

in the downstream prediction task. Table 4 shows the comparison of the SELFIES-TED$_{\text{large}}$ in comparison with the SMILES-based models across all 12 target properties of the QM9 dataset. The SELFIES-TED$_{\text{large}}$ model has an overall better mean absolute error compared to the other models.

| QM9 Properties | MolFormer-XL | ChemBERTa | MolGen$_{\text{large}}$ | SELFIES-TED$_{\text{large}}$ |
|---|---|---|---|---|
| $\alpha$ | 1.0749 | 0.8510 | 1.1711 | 0.6686 |
| $C_v$ | 0.5078 | 0.4234 | 0.4362 | 0.4377 |
| $G$ | 2.995 | 4.1295 | 7.4269 | 2.2922 |
| gap | 0.0084 | 0.0052 | 0.0326 | 0.0084 |
| H | 2.6831 | 4.0853 | 7.4263 | 2.7049 |
| $\epsilon_{\text{homo}}$ | 0.0054 | 0.0044 | 0.0193 | 0.0054 |
| $\epsilon_{\text{lumo}}$ | 0.0065 | 0.0041 | 0.0389 | 0.0071 |
| $\mu$ | 0.5981 | 0.4659 | 0.6735 | 0.6223 |
| $\langle R^2 \rangle$ | 46.384 | 86.150 | 118.80 | 38.832 |
| $U_0$ | 3.2735 | 3.9811 | 7.4266 | 2.9195 |
| $U$ | 3.2791 | 4.3768 | 7.4264 | 2.6551 |
| ZPVE | 0.0038 | 0.0023 | 0.0228 | 0.0032 |
| **Overall MAE** | 5.0691 | 8.7067 | 12.575 | **4.263** |

Table 4: A detailed comparison across different measures in the QM9 dataset.

From the evaluations on both classification and regression tasks, we can observe that the SELFIES-TED model outperforms existing models in most of the tasks. The comparison between SELFIES-TED$_{\text{small}}$ and SELFIES-TED$_{\text{large}}$ further highlights the effect of model scale. SELFIES-TED$_{\text{large}}$ consistently outperforms SELFIES-TED$_{\text{small}}$, attributed to its larger model parameter count and the increased diversity of its training data, enabling it to better capture complex molecular features. Despite this, SELFIES-TED$_{\text{small}}$ remains a competitive option for applications with limited computational resources, offering a balance between efficiency and performance. This scalability underscores the flexibility of the SELFIES-TED framework in addressing diverse computational constraints while maintaining high performance. The superior performance of SELFIES-TED is attributed to its encoder-decoder architecture, which when trained on SELFIES representations with a denoising objective, ensures robust and valid molecular embeddings. Unlike SELFormer which uses an encoder-only architecture, the encoder-decoder structure of SELFIES-TED enables more expressive and robust representations, further enhancing its predictive accuracy across tasks. Furthermore, while MolGen also uses an encoder-decoder architecture, the improved performance of SELFIES-TED is driven by its large-scale training, with SELFIES-TED$_{\text{small}}$ and SELFIES-TED$_{\text{large}}$ trained on 8 billion and 1 billion samples, respectively, highlighting the critical role of extensive pretraining in achieving state-of-the-art results.

**Preliminary Evaluation of the Proposed Multi-View Representation Approach**: In addition to the above evaluations, we conduct a preliminary evaluation of the Multi-View Representation (MVR) approach described in Section 3. For this we choose three regression and two classification tasks based on datasets with few samples. For each molecule in the dataset used for evaluation we generate 4 alternate SELFIES strings, one of which is the canonical set. Thus $k = 5$ including the original dataset. The latent representation of the molecules for each set is extracted using the encoders of SELFIES-TED$_{\text{large}}$ model. The extracted latent representations are concatenated in combinations of $k = 2, 3, 4, 5$ and a greedy selection method is applied to select the best combinations to form the new enriched latent representation as detailed in Section 3. The corresponding results are reported in Table 5. As seen from the table, the MVR approach has an improved performance thus confirming that the latent representations of the alternate forms of the molecules capture additional information, which helps in improving the quality of the resultant latent representation and thus improving the evaluation score. Notably, concatenating all $k = 5$ representations does not necessarily

| Model | ESOL $\downarrow$ | FreeSolv $\downarrow$ | Lipophilicity $\downarrow$ | ClinTox $\uparrow$ | BACE $\uparrow$ |
|---|---|---|---|---|---|
| SELFIES-TED | 0.454 | 1.147 | 0.672 | 96.90 | 89.30 |
| SELFIES-TED (w/ canonical) | 0.406 | 1.251 | 0.671 | 93.91 | 89.77 |
| SELFIES-TED w/ MVR (k=2) | **0.373** | 1.136 | **0.648** | 90.95 | 89.58 |
| SELFIES-TED w/ MVR (k=3) | 0.378 | **1.123** | 0.661 | **97.51** | **90.02** |
| SELFIES-TED w/ MVR (k=4) | 0.387 | 1.156 | 0.675 | 85.27 | 88.99 |
| SELFIES-TED w/ MVR (k=5) | 0.396 | 1.279 | 0.688 | 86.33 | 89.16 |

Table 5: Evaluation of Multi-View Representation on MoleculeNet benchmark datasets

yield the best results, highlighting the importance of selective combination. Given $k = 5$, there are 31 possible combinations, necessitating the use of the greedy selection strategy. While $k$ was fixed in this preliminary analysis, the number of alternate representations ($k$) is a hyperparameter, and future work may explore more efficient methods for optimizing it.

## 4.2 MOLECULE GENERATION TASK

The SELFIES-TED model is an encoder-decoder architecture, making it not only capable of providing robust molecular representations but also adept at generating molecules. In this section, we evaluate the generative performance of our best-performing model, SELFIES-TED$_{large}$ (hereafter referred to as SELFIES-TED), to assess its capabilities in molecule generation tasks. Given the infinitely large and unexplored chemical space, it is crucial for a molecular generative model to understand molecular grammar and rules, ensuring the generation of novel and valid molecules. As a preliminary analysis, we first evaluate the SELFIES-TED model's ability to generate valid molecules upon randomly sampling the latent space. This gives us an insight of the learned representations. Previous works such as Reidenbach et al. (2022); Noutahi et al. (2024) have explored molecule generation by perturbating the latent space and evaluated model performances based on validity, uniqueness and novelty scores.

For this molecule generation task, we randomly select 10,000 samples curated from Zinc and Pub-Chem dataset to form the reference set, and obtain their latent representations. We then perturb the latent representations by applying uniform random noise, thus creating random samples from the latent representation. These are then fed to the SELFIES-TED decoder to generate molecules from the sampled latent space. We evaluate the molecules generated by the decoder based on standard metrics. This evaluation helps us understand the model's proficiency in producing diverse and valid molecular structures. The metrics we use in this analysis are validity, uniqueness, novelty, internal diversity and FCD score. These metrics are evaluated using the MOSES package (Polykovskiy et al. (2020)). Validity measures how well the model has learned the molecular grammar and rules. Novelty gives a measure of the model's capability to generate unique molecules that are not in the reference set while uniqueness indicates if the model is prone to repetitive generation of same molecules and is indicative of the level of distribution learnt by the model. Internal Diversity Score (IntDiv$_p$) measures the diversity of the generated molecules and the tendency of the model to generate similar structures repetitively. The IntDiv$_1$ (p = 1) and IntDiv$_2$ (p = 2) scores are reported. Similarly, the Fréchet ChemNet Distance (FCD) (Preuer et al. (2018)) score measures the similarity between the distributions of generated and real molecules. It compares molecular features, such as chemical properties, by calculating the distance between embeddings of both sets. A lower FCD score indicates that the latent space distribution of the generated molecules are more similar to real ones, making it a useful metric for evaluating the quality of generated molecular structures.

The metric scores are presented in Table 6. The metrics for CharRNN, VAE, AAE, LatentGAN, JT-VAE and MolGPT are values reported from Bagal et al. (2021). From the results, we can observe that the SELFIES-TED model is equally performant in generating unique, valid, and novel molecules with the high internal diversity, and low FCD score, thus confirming its effectiveness in generating molecules of varying structures and quality compared to similar baseline methods. We also calculated common properties such as QED (Quantitative Estimate of Drug-likeness), SA (Synthetic Accessibility) score, logP (solubility coefficient) and molecular weight, using RDKit. Figure 5 show the density plots of the generated molecules and the reference set. The close alignment of the density curves across all four properties suggests that the SELFIES-TED model is effective in generating molecules that are similar to the reference molecules in terms of drug-likeness,

| Models | Validity ↑ | unique@10K ↑ | Novelty ↑ | IntDiv$_1$ ↑ | IntDiv$_2$ ↑ | FCD ↓ |
|---|---|---|---|---|---|---|
| CharRNN | 0.975 | 0.999 | 0.842 | 0.856 | 0.85 | 0.073 |
| VAE | 0.977 | 0.998 | 0.695 | 0.856 | 0.85 | 0.099 |
| AAE | 0.937 | 0.997 | 0.793 | 0.856 | 0.85 | 0.555 |
| LatentGAN | 0.897 | 0.997 | 0.949 | 0.857 | 0.85 | 0.297 |
| JT-VAE | 1.0 | 0.999 | 0.914 | 0.855 | 0.849 | 0.395 |
| MolGPT | 0.994 | 1.0 | 0.797 | 0.857 | 0.851 | 0.297 |
| SELFIES-TED | 1.0 | 0.991 | 1.0 | 0.867 | 0.862 | 0.206 |

Table 6: Comparison of molecular generative models across key evaluation metrics.

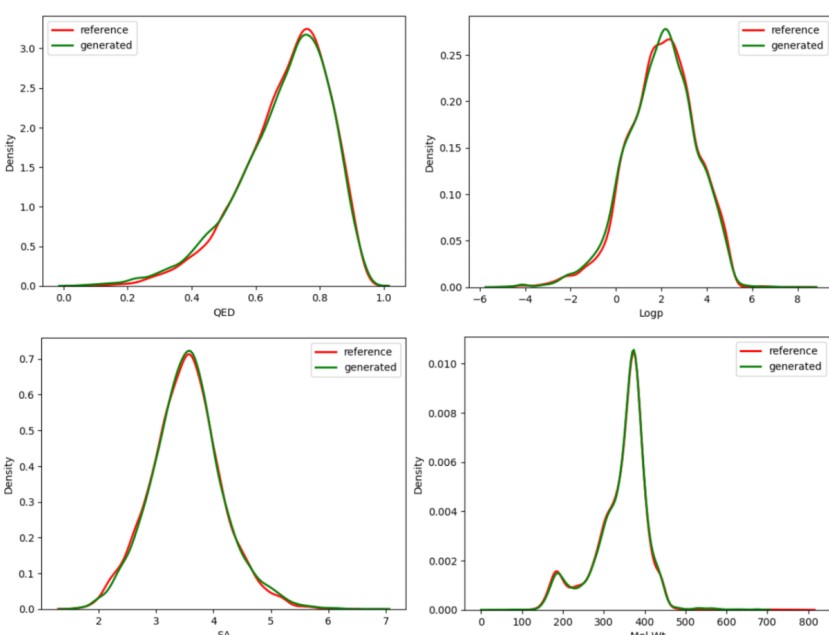

Figure 5: Density plots showing the properties of the model generated molecules

lipophilicity, synthetic accessibility, and molecular weight. Figure 6 shows the t-SNE plot of the latent space distribution of the generated molecules color coded based on their molecular weights. The clustering and smooth transition of colors across the space suggest that the latent space preserves certain chemical properties, such as molecular weight, indicating that the model may have learned meaningful patterns related to molecular composition.

Evaluating a model's ability to generate novel molecules that optimize key properties is also crucial for advancing applications in drug discovery, materials science, and beyond. A robust generative model should not only propose chemically valid and novel molecules but also improve critical attributes, such as drug-likeness or synthetic accessibility, while retaining the core structure of a reference molecule. This capability allows researchers to efficiently identify and refine high-potential candidates for further experimental validation, significantly accelerating the discovery process. To test this aspect of the SELFIES-TED model, we evaluated its ability to generate new molecules by exploring the latent space around a given reference molecule. The goal was to generate molecules with improved properties, specifically higher QED (quantitative estimate of drug-likeness) and lower SA (synthetic accessibility) scores compared to the reference molecule. The SELFIES-TED model successfully generated new molecules with these desired properties while maintaining a high Tanimoto similarity to the reference molecule, indicating that the core structural features were preserved. Examples of the molecules generated by SELFIES-TED, along with their corresponding properties and Tanimoto similarity scores, are presented in Figure 7. The regions of the generated molecules that differ from the reference structure are highlighted in yellow, illustrating how the model modifies substructures to optimize target properties. These results underscore the model's ability to navigate the chemical latent space, making it a potential tool for tasks requiring the optimization of molecular properties while preserving the essential scaffold.

## 5 CONCLUSION

This paper introduces SELFIES-TED, an encoder-decoder transformer model specifically designed to learn effective representations of the chemical space. By leveraging SELFIES strings during training, SELFIES-TED ensures molecular validity, enhancing the reliability and robustness of its molecular representations. The model's performance was thoroughly evaluated using benchmark classification and regression tasks from MoleculeNet, where it achieved state-of-the-art results in most cases. Beyond the standard downstream tasks, this work extends the exploration of molecular latent representations by incorporating a multi-view representation approach, enriching the diversity and depth of the model's chemical understanding. Additionally, the model's capability to generate

novel molecules was demonstrated by both random sampling from the latent space and optimizing molecular designs to achieve more desirable properties, such as improved drug-likeness and synthetic accessibility. Preliminary analysis highlights that SELFIES-TED is not only capable of generating valid and novel molecules but also exhibits strong structural diversity. These findings indicate the model's potential to significantly advance molecular discovery and optimization, offering a promising tool for drug development and other chemistry-driven fields.

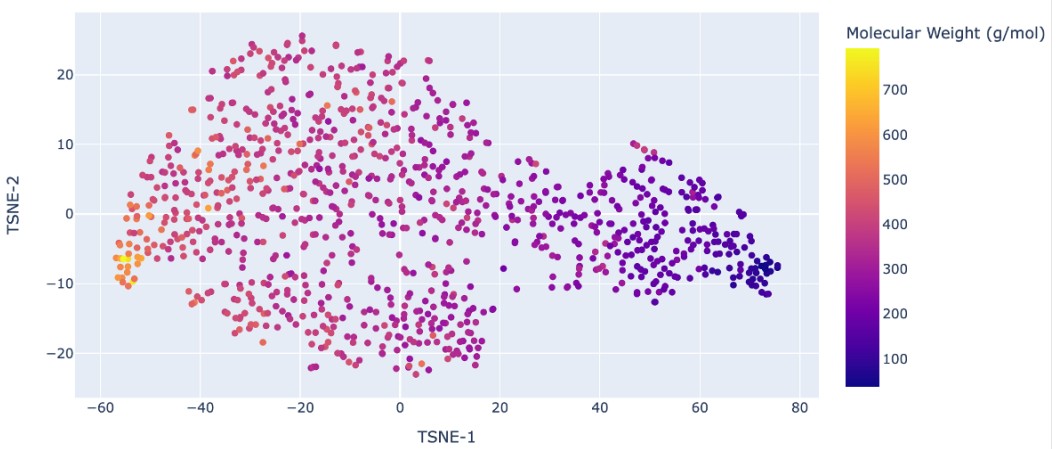

Figure 6: t-SNE visualization of the latent space distribution of generated molecules

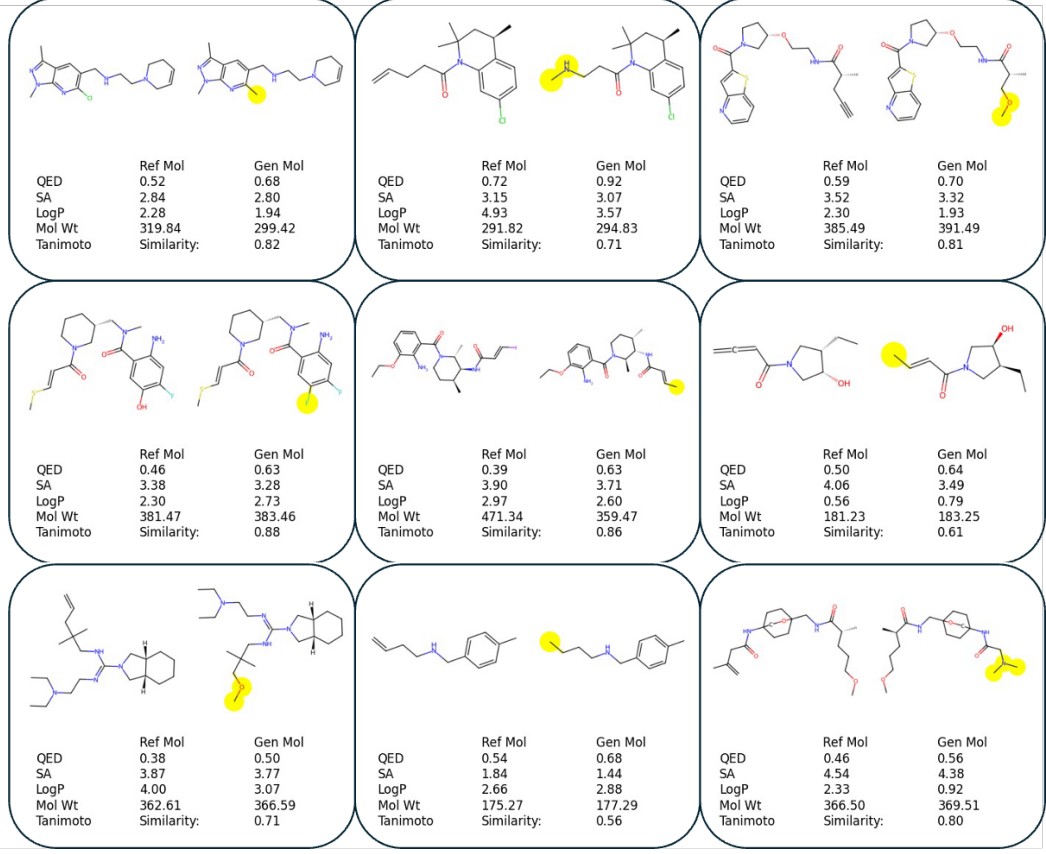

Figure 7: Molecules generated by SELFIES-TED with improved QED values compared to a given reference molecule

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
