# OpenReview forum: "SELFIES-TED : A Robust Transformer Model for Molecular Representation using SELFIES"
_ICLR.cc/2025/Conference — Submitted to ICLR 2025_

### Official Review · Reviewer_xQbW · 2024-10-25

**Soundness:** 1
**Presentation:** 2
**Contribution:** 1
**Rating:** 1
**Confidence:** 5

**Summary:**

SELFIES-TED introduces a transformer trained on SELFIES strings for improved molecule property prediction. SELFIES-TED uses a BART backbone to learn a molecule representation while also being able to generate novel molecules. SELFIES-TED has 354M parameters and was trained on 1 billion molecules from zinc-22, applying smiles enumeration. The authors also introduce a multi-view representation where multiple valid selfie representations are aggregated for one improved latent vector.

**Strengths:**

Overall, the paper is short and succinct. The multi-view representation is an interesting approach to leveraging existing data and representations to improve property prediction results. The results show the model can improve property prediction for certain tasks.

**Weaknesses:**

The paper is quite sparse in its details. Outside of the model name, archetype, and size, no details are given for the model training and benchmarking, making it quite difficult to understand the scientific contributions. Optimal hyperparameters are referred to but not shared. I think there is a strong possibility for this to be a meaningful work, but a significant addition of information and experiments is needed to understand the contribution.

There is little novelty in taking the same training data as prior methods and swapping out the transformer backbone to train a larger model, especially when the BART architecture has been explored with selfies before [1] and was not cited.

The Multi-view representation is interesting but not specific to SELFIES or SELFIES-TED and should be properly ablated by comparing it against prior property prediction methods. To understand the comparison of SELFIES vs. SMILES and BERT vs. BART, it would be important to have training ablations, even if on a smaller scale.

There are also several claims on the improvements SELFIES yield over SMILES but no experiments are given to substantiate those claims. There have been several works exploring these claims, including [2], which argue that invalid SMILES are enriched among low-likelihood samples from chemical language models. No discussion on this area of work is provided when central to the primary contribution.

- Given the paper is focused on introducing SELFIES-TED as a novel model the training and inference details as well as ablations are necessary as can be seen in section 4 of SELFformer of a similar method.
- Molformer -XL is at 47M params, SELFformer 87M, and UniMol 47M, yet only one size of SELFIES-TED is reported at 354M. Given the difference between SELFformer and SELFIES-TED is RoBERTa vs. BART, significant ablations are necessary to understand the resulting benefit, and it is worth a 4x increase in model size at a minimum.
- Prior methods have also explored BART for SMILES and SELFIES and explored the issue with a variable length representation and are not cited nor compared against [3, 1].
- No training information is given about any hyperparameters for the LLM or the classifier and regression models trained for the benchmarks.
- The molecule generation benchmarks are quite sparse, with all baselines taken from MolGPT, which was published three years ago. Large SELFIES-based models like SAFE-GPT [4] also include several other prior SELFIES-based models for molecule generation.




[1] https://arxiv.org/pdf/2301.11259

[2] Invalid SMILES are beneficial rather than detrimental to chemical language models https://www.nature.com/articles/s42256-024-00821-x

[3] https://arxiv.org/pdf/2208.09016

[4] https://arxiv.org/abs/2310.10773

**Questions:**

- How was SELFIES-TED trained?
- How does the classification performance depend on the model size?
- What happens if you apply the multiview representation to similar SELFIES and SMILES models that also rely on enumeration for training?
- Given that BART uses a variable-length encoding scheme like all prior BERT and BART-based architectures, how is the input to the XGBOOST classifier obtained when a latent vector of size 1024 is produced for each token, is it a average over the sequence length?
  - Are the prior methods compared against using the same latent vector size?
- How does SELFIES-TED compare to Morgan Fingerprints?
- What would happen if the multi-view representation were used with ML and classical methods?

---

> ### Author Response · Authors · 2024-11-30
>
> We thank the reviewers for their comments and feedback.
> - We have revised the paper with details on model training and the hyperparameter tuning.
> - We thank the reviewer for pointing us to the work MolGen-large (https://arxiv.org/pdf/2301.11259). We have included this work in the evaluations based on the model available on huggingFace.
> - We agree that the Multi-view Representation (MVR) method is not exclusive to SELFIES or SELFIES-TED but is a versatile approach applicable to models where SMILES enumeration is feasible. Our paper introduces MVR as a general framework and demonstrates its effectiveness with SELFIES-TED because this model is central to our study. As shown in Table 6, MVR enhances the performance of SELFIES-TED, further validating the concept. Additionally, from the results in Tables 2, 3, and 4, SELFIES-TED consistently outperforms other models across the majority of tasks, showcasing its robustness. By combining SELFIES-TED with MVR, we achieve even better results, underscoring the value of the proposed approach.
> - We have added evaluations results of MorganFingerprint to compare with the proposed SELFIES-TED model
> - MVR can be used with ML methods. For Regression tasks such as esol, freesolv, SVR is used (Table 5).
> - The latent vector is obtained by averaging over the sequence length. Thus the input to the XGBoost classifier is a (n, 1024) vector, where n is the number of samples
> - To show the effect of model scaling, we have included results of a smaller SELFIES-TED model (2.2M parameter) that was trained on 8B samples.
> - Thank you for pointing us to prior works such as [3] and SAFE-GPT [4] that are based on SELFIES. We have cited these works in the revised paper.  Both papers [3] and [4] evaluate molecule generation on metrics such as novelty, uniqueness and diversity. From the results shown in Table 3 in ref [1], our proposed model outperforms the model in [1] and SAFE-GPT-20M, and comparable with the diversity score of SAFE-GPT, which  uses a different training dataset that includes non drug-like and challenging molecules. SELFIES-TED as described in the paper is trained on ZINC-22 and PubChem dataset, which is primarily drug-like molecules.

---

> > ### Comment · Reviewer_xQbW · 2024-11-30
> >
> > Overall, the majority of my concerns are not addressed. What is primarily lacking is a discussion and explanation of how results are obtained and their significance. A deeper discussion of how the model was trained, how the data was curated, and why the architecture is the way it is is crucial to understanding any scientific significance beyond better benchmark scores. From reading this paper, it is impossible to tell what led to the improvement and how to replicate or build off that.
> >
> > - Where are the hyperparameter tuning details? Only single values are provided.
> > - The provided model ablations are not meaningful and quite confusing. The small model is given 8x more data but only 2 layers? There should be at least somewhat of a scaling analysis that leads to the 358M especially due to the size increase compared to prior models. The 2-layer model seems unrealistic as all prior methods, even non-LLM models, have more than 2 layers. Also, data is not kept the same, which voids a direct comparison.
> >     - The small model is a 2.2M parameter model with 2 encoder-decoder layers and 4 attention heads, pretrained with 8B samples
> >     - The large model is a 358M parameter model with 12 encoder-decoder layers and 16 attention heads, pretrained with 1B samples.
> > - Where is the discussion of how the Morgan Fingerprint results were obtained?
> > - If MVR is being proposed as a general framework, there needs to be evidence to substantiate that claim. Currently, there is none, and still, the paper is lacking any and all critical ablations.
> >
> > Overall, the paper is still majorly unfinished. It lacks explanations of the method and experiments. It has uninformative experiments and lacks technical novelties as many prior works that were initially missed and now added are not properly ablated.
> >
> > I maintain my stance on strong rejection. I recommend that the authors do a proper ablation study and take the time to write a self-sustaining paper upon resubmission.

---

### Official Review · Reviewer_kDCp · 2024-10-30

**Soundness:** 2
**Presentation:** 3
**Contribution:** 2
**Rating:** 6
**Confidence:** 2

**Summary:**

This work introduces SELFIES-TED, a transformer-based encoder-decoder model that uses SELFIES to learn molecular representations. The model according to authors achives very high competitive performance on various benchmarks. The authors also propose a Multi-View Representation approach that leverages multiple representations of the same molecule to improve prediction accuracy. Authors have repurposed the BART model and trained it using PubChem+Zinc 22. In order to improve the prediction accuracy the authors used a method to create multiple SMILES representations for a given molecule and then generate the SELFIES from those generated strings and used it in MVR.

**Strengths:**

The authors introduce a Multi-View Representation (MVR) approach, which could potentially help one improve training when there are unbalanced data points, especially in chemistry. The bidirectional approach to molecular representation learning differs from traditional encoder-only models. The model looks promising when we look into the benchmark results and also shows promising results on property prediction and molecule generation. The paper looks well-written and formulated the figures are clear. Methods and architecture are well-explained. The paper also provides insights into the latent space organization of molecular features. Also, it is great to see such work is completely open-sourced. which is highly commendable.

**Weaknesses:**

The authors do not state the frequency of their SELFIES lengths or how big of a molecule they could generate with the trained model.  Moreover, they also mention that they randomly sampled 10000 molecules from the training set to generate new molecules to understand how diverse the molecules generated are. To create a diverse dataset the authors should have picked a diverse set instead of a random dataset and then generated molecules which could ensure how well the model covers the whole chemical space.  Which is not clear in the paper. The MVR approach could benefit from a more theoretical analysis of why certain combinations of views work better. The paper does not clearly explain why this model works better than other models which are based on SELFIES. Also, the authors should have taken a bit more effort to make the paper ready for a double-blind review a simple Google search leads me to the IBM huggingface portal.
The authors do not introduce any new model but rather repurpose an already existing model it should be mentioned how the model was chosen.

**Questions:**

Are there any limitations in the generated length of the SELFIES string?
Figure 6 should have the training dataset chemical space and be compared it with the generated molecules' chemical space.
The MVR idea is interesting but I have a question for the authors did they try to parse the input data always through RDKit create canonical SMILES and then generate and train the model how well does the model compare to the original?
The authors should reduce the use of unnecessary words such as "state of the art", "enhancing the reliability", "enriched" and so forth, which can be seen throughout the paper.
The authors should also discuss the limitations of using this model.

---

### Official Review · Reviewer_6rtt · 2024-11-03

**Soundness:** 2
**Presentation:** 1
**Contribution:** 1
**Rating:** 1
**Confidence:** 4

**Summary:**

The paper trains an encoder decoder transformer on SELFIES string representations of small molcules on large unfiltered datasets. For evaluation it evaluates prediction and generation tasks. It trains predictors on top of the latent representations and checks their predictions in various benchmarks. It generates molecules unconditionally and evaluates their distributional properties.

**Strengths:**

1. Interesting histograms of generated and reference molecule's synthetic acessibility and QED scores.
2. Reasonable experiments on moleculenet property predictions

**Weaknesses:**

1. Missing information: It is not sufficiently explained how the transformer is trained. Is there always only 15% of the sequence masked? Is X_{corrupt} a 15% masked SELFIE or is Y_{<t} the 15% masked SELFIE? If X_corrupt is the 15% masked selfie and Y_{<t} experiences all masking ratios but autoregressively from left to right, then why would the loss not be minimized by just copying over the information from X_{corrupt}?
2. Missing information: How do you compute novelty? What is the reference set and what is the similarity measure.
3. The fact that multiple smiles describe the same molecule is a bug not a feature. You introduce Multi View representation as a workaround, but this would not be necessary if one simply employs a representation learning model that encodes molecules instead of ambiguous representations of molecules such as SMILES or SELFIES. (Even when using selfies/Smiles, could one not use a canonicalized smiles/selfie version instead of the ambiguous one? I know this exists for SMILES and would guess that it is also possible to construct for SELFIES.)

### Experiments:
1. No evaluation of correctly sampling the generative model: The paper evaluates small molecule generation by embedding existing molecules and perturbing their latents instead of sampling the transformer autoregressively.
2. The histograms in Figure 5 and the FCD are computed between the set of small molecules that is used to sample the "generated" small molecules around and the set of "generated" small molecules. If the perturbation noise simply is very small (e.g. as small as in Figure 7), then the "generated" small molecules will be (almost) identical to the input/refernce molecules - we did not generate anything new at all and the distributional metrics would all look very good. The only metric that would suffer from this is novelty. However, it is not explained w.r.t. which set novelty is computed and how novelty is computed at all. A typical novelty score would be reporting the maximum tanimoto similarity where 1 is the worst possible score. In the provided table the score for novelty for SMILES-TED is 1.
3. For the QM9 evaluations only SMILES/SELFIE embedding transformer based models are considered instead of GNNs or other regression models that predict from the molecule.


The paper does not seem finished: it is missing explanations of the method and experiments, has uninformative experiments and no useful technical novelties. I do not think that it would be of value to any readers and recommend strong rejection.

**Questions:**

1. How do you compute novelty?
2. What is the magnitude of the noise that is added to the embeddings for Table 6 and Figure 5 compared to the magnitude of the noise added for figure 7.
3. How do you obtain the fixed size single vector for every SELFIE when making TSNE plots - do you sum the latent representations of every token?

---

> ### Author Response · Authors · 2024-11-27
>
> We thank the reviewer for their feedback. We have revised the paper clarifying the concerns raised.
>
> *** Denoising objective  ***
> - We have updated the paper with details on the model training and tokenization. The model is an encoder-decoder model with a denoising objective. Given an original sequence $Y$, we prepare the input by applying mask to 15% of the tokens, resulting in a noisy corrupt sequence $X_{corrupt}$ .  The encoder processes this corrupted input sequence (\(X_{\text{corrupt}}\)), and the decoder is trained to autoregressively predict the next token in the original sequence (\(Y\)), conditioned on the corrupted sequence and the previously decoded tokens (\(Y_{<t}\)). Specifically, the decoder learns to recover the original sequence by predicting each token in the correct order, rather than directly copying over the corrupted input.
>
> *** Multiple SMILES for the Same Molecule is NOT a Bug ***
>
> The ability to represent a molecule with multiple SMILES strings is an inherent feature of the SMILES notation, not a limitation or bug. This flexibility arises because the SMILES representation depends on the starting atom and the traversal order within the molecular graph. An analogy can be drawn to viewing an object from different perspectives (e.g., front view, top view, side view). Each view provides complementary information about the object, contributing to a more comprehensive understanding.
>
> Similarly, we propose in this paper that representing a molecule with multiple SMILES or SELFIES strings—arising from variations in starting atoms in a ring, traversal orders, or chirality—can provide additional information about the molecule. As shown in Figure 3, the latent representations of these enumerated SMILES form meaningful clusters. By concatenating these representations, we construct an enriched feature vector that captures more nuanced molecular features. This enriched representation consistently outperforms using a single canonical representation, as demonstrated by the improved results in Table 4. Our findings validate that leveraging multiple SMILES/SELFIES representations enhances molecular representation and downstream task performance.
>
>
> *** Novelty and similarity measure ***
>
> - The paper has two similarity measures - FCD and Tanimoto similarity.
>
> - FCD (Fréchet ChemNet Distance) is a metric used to evaluate the similarity between two sets of molecules in terms of their distribution in chemical space. A lower FCD value indicates that the generated molecules closely match the reference molecules in their chemical properties, suggesting that the generative model effectively captures the underlying distribution of the reference dataset.
> https://pubs.acs.org/doi/10.1021/acs.jcim.8b00234
>
> - Figure 7 measures the Tanimoto similarity between the reference molecule and generated molecule to measure the structural similarities.
>
> - The reference set is formed by randomly selecting 10,000 samples curated from Zinc and PubChem dataset. The reference set serves as the initial set of molecules whose latent representation is perturbed for generation of new molecules.
>
> - Novelty is computed as the fraction of generated molecules that do not exist in the reference set. We would like to clarify that novelty score is not related to the tanimoto similarity.
>
> *** Sampling in generative task ***
> - As described in the paper, for Figure 5 and Table 6, the evaluation involves sampling random points in the latent space by applying uniform random noise to the latent representations of 10,000 molecules from a curated reference set. Figure 7 uses small perturbations to illustrate interpolation dynamics, while Figures 5 and 6 use larger noise magnitudes to explore random sampling of the chemical latent space.
>
> - In material or drug discovery, it is common to generate a new molecule by starting with a given initial reference molecule and search the chemical latent space by small perturbations until we find a molecule that is similar or close to the initial molecule but with certain desired properties (for example see https://arxiv.org/pdf/2208.09016) .Figure 7 is an example of this where given a reference molecule, we search the chemical latent space to find a molecule that is similar (high Tanimoto similarity) and with certain desired property (here lower SA and higher QED compared to reference molecule).
>
> - The encoder output (fixed length latent representation vector) is obtained by averaging over the sequence length, which results in a n x 1024 vector, where n is the number of samples.
>
> We have revised the paper with a few additional evaluations clarifying the concerns of the reviewers. We believe the current paper is complete, provides novel technical contributions, and offers valuable insights through robust methodology and comprehensive experiments. We respectfully request the reviewer to reconsider their assessment in light of these clarifications.

---

> > ### Comment · Reviewer_6rtt · 2024-12-01
> > **Response by Reviewer**
> >
> > Several of my concerns remain ignored or incompletely addressed. I reconsidered my assessment and maintain that the paper is incomplete and that the experiments are uninformative.
> >
> > For instance, the rebuttal clarified that the paper does not train a correct generative model that targets a particular distribution of interest and refuses to compare with this.
> >
> > The revised version of the paper does not contain the missing information that I pointed out as necessary, such as how the novelty metric is computed.

---

### Official Review · Reviewer_S26N · 2024-11-04

**Soundness:** 3
**Presentation:** 2
**Contribution:** 3
**Rating:** 6
**Confidence:** 4

**Summary:**

SELFIES-TED, a transformer-based model for molecular representation, property prediction and generation that utilizes SELFIES, was proposed in the paper. The model uses a robust and unambiguous molecular string representation, compared to traditional SMILES. An encoder-decoder architecture inspired by BART was introduced and can capture complex molecular relationships and features. The MVR approach further enhances the quality of the learned representations. The model was evaluated on both molecular property prediction and generation tasks. The results are compared with several state-of-the-art works. SELFIES-TED is more accurate and the generated molecules have better validity and novelty.

**Strengths:**

1. The proposed SELFIES-TED ensures the syntactic and semantic validity representation of molecules, This significantly reduces the chances of learning invalid molecular representations and makes the model more robust compared to those relying solely on SMILES.
2. Inspired by BART, the Encoder-Decoder transformer models are valuable for generating molecules. The MVR approach also expands the latent representation and generates multiple SELFIES strings, which significantly improve the quality of proposed models.
2. The model was compared with several graph-based, geometry-based, and text-based models. It achieves state-of-the-art results in many tasks, showing its potential to advance molecular representation learning. Specifically, the proposed model outperforms related works on several molecular property prediction and generation tasks.
3. The validity, uniqueness, and novelty of the proposed molecules are significantly better than previous works. The distribution of generated molecules is also similar to reference molecules.

**Weaknesses:**

1. In Figure 3, there are only 10 different molecules. Some latent representations of the green mols are surrounded by red molecules. Will this limit the model if there are a lot of molecules in the dataset? And, how about if the dataset is very small but the molecules are very similar?
2. Could you please highlight the difference between Ref Mol and Gen Mol in Figure 7? The molecules have very similar 2D graphs.
3. The platform that trains and evaluates the models is not introduced. The overhead and limitation of SELFIES-TED are not well explained.

**Questions:**

1. In Figure 5, why the density of logp is not as the others? Does this mean that the model not doing well for logP?
2. How many representations are needed in MVR? Does it mean that the model is n times slower if n representations are generated?
3. Why do you use the greedy selection method in the MVR? Will this lead to overfitting?
4. Do the canonical SMILES have some privilege so that they are more likely to be selected?

---

> ### Author Response · Authors · 2024-11-27
>
> We thank the reviewer for their thoughtful feedback and evaluation.  We have revised the paper accordingly.
>
> - Overlapping of clusters Figure 3 : Overlapping of clusters can be expected, as different molecules can share structural similarities. This reflects the complexity of chemical space, where structurally or functionally similar molecules may cluster together. The results presented in Table 4  demonstrate the MVR approach with SELFIES-TED's effectiveness across datasets with varying sizes, such as FreeSolv (642 samples) and Lipophilicity (4,200 samples) and the performance is consistent.
>
> - Number of representations ($k$): Figure 3 illustrates $k=100$ alternate enumerations of the molecules to visualize the spread of the molecular latent representations. While this visualization helps highlight the diversity of representations, for model evaluation, we recommend smaller values of \(k\). Increasing \(k\) beyond a certain point does not necessarily improve performance, as it can lead to the curse of dimensionality, dilute the relevance of the latent space, and increase computational overhead.
>
> - Greedy selection method:  As presented in Table 4, concatenating all k = 5 representations does not necessarily yield the best results, highlighting the importance of selective combination. It does not lead to overfitting since we use greedy selection to find the optimum number of representations ($k$) and which combination of $k$ alternative representations yield best results.
>
> - Role of canonical representations: In the revised version, we added canonical representations evaluations as one of the possible representations in Table 4. The canonical representations do not always outperform other combinations and are not guaranteed to improve performance. Some of the best-performing MVR configurations $(k=2,3,4$) does not include the canonical representation, indicating that the canonical representation does not have any privilege over other representations.
>
> - The difference in between the reference and generated molecules are highlighted in yellow in Figure 7 and pre-training details are updated in the revised draft.
>
> - logp density plot : Figure 5  shows the density distribution plots of the reference and generated molecules. The generated molecules exhibit distributions closely aligned with those of the reference molecules across all four properties. This indicates that the SELFIES-TED model effectively captures the underlying property distributions of the reference dataset and generates molecules with similar characteristics.  The slight misalignment in the LogP distribution does not necessarily imply that the model is not doing well with logP. LogP is a property heavily influenced by specific molecular substructures, such as hydrophobic or hydrophilic groups. So small structural changes can lead to significant shifts in LogP values.

---

### Meta-Review · Area_Chair_x911 · 2024-12-18

**Metareview:**

The authors tested SELFIES-TED, leveraging the robustness of SELFIES molecular representation and the transformer encoder-decoder architecture for molecular property prediction. Results are presented with benchmark datasets, including QM9.

Besides the concern on limited methodological contributions, most of the reviewers consider the current version of the submission is not ready for publication. The authors shall consider improving the presentation by providing clear technical detail description and preforming more comprehensive evaluation experiments.

**Additional Comments On Reviewer Discussion:**

After the rebuttal discussions, reviewers were not convinced the submission has sufficient methodological contributions or strong empirical evidence to support the significance of the presented work.

---

### Decision · Program_Chairs · 2025-01-22

Reject